# WISP1 Is Involved in the Pathogenesis of Kashin-Beck Disease via the Autophagy Pathway

**DOI:** 10.3390/ijms242216037

**Published:** 2023-11-07

**Authors:** Ping Li, Bolun Cheng, Yao Yao, Wenxing Yu, Li Liu, Shiqiang Cheng, Lu Zhang, Mei Ma, Xin Qi, Chujun Liang, Xiaomeng Chu, Jing Ye, Shiquan Sun, Yumeng Jia, Xiong Guo, Yan Wen, Feng Zhang

**Affiliations:** 1Key Laboratory of Trace Elements and Endemic Disease of National Health Commission of the People’s Republic of China, School of Public Health, Health Science Center, Xi’an Jiaotong University, No. 76 Yanta West Road, Xi’an 710061, China; doushiyu@163.com (P.L.); cblbs1@stu.xjtu.edu.cn (B.C.); yyxjtu@xjtufh.edu.cn (Y.Y.); liuli0624@stu.xjtu.edu.cn (L.L.); chengsq0701@stu.xjtu.edu.cn (S.C.); snnuyang@snnu.edu.cn (L.Z.); captainooo@stu.xjtu.edu.cn (M.M.); xinqi0702@xjtufh.edu.cn (X.Q.); 13259880170@163.com (C.L.); xiaomengchu.zh@gmail.com (X.C.); applejuice@stu.xjtu.edu.cn (J.Y.); sqsunsph@mail.xjtu.edu.cn (S.S.); jiayumeng@mail.xjtu.edu.cn (Y.J.); guox@mail.xjtu.edu.cn (X.G.); 2Department of Joint Surgery, Xi’an Honghui Hospital, Health Science Center, Xi’an Jiaotong University, Xi’an 710054, China; yuwenxing110@163.com

**Keywords:** Kashin-Beck disease, WISP1, autophagy, chondrocytes

## Abstract

Objective: Kashin-Beck disease (KBD) is a kind of endemic and chronic osteochondropathy in China. This study aims to explore the functional relevance and potential mechanism of Wnt-inducible signaling pathway protein 1 (WISP1) in the pathogenesis of KBD. Design: KBD and control cartilage specimens were collected for tissue section observation and primary chondrocyte culture. Firstly, the morphological and histopathological observations were made under a light and electron microscope. Then, the expression levels of WISP1 as well as molecular markers related to the autophagy pathway and extracellular matrix (ECM) synthesis were detected in KBD and control chondrocytes by qRT-PCR, Western blot, and immunohistochemistry. Furthermore, the lentiviral transfection technique was applied to make a WISP1 knockdown cell model based on KBD chondrocytes. In vitro intervention experiments were conducted on the C28/I2 human chondrocyte cell line using human recombinant WISP1 (rWISP1). Results: The results showed that the autolysosome appeared in the KBD chondrocytes. The expression of WISP1 was significantly higher in KBD chondrocytes. Additionally, T-2 toxin, a risk factor for KBD onset, could up-regulate the expression of WISP1 in C28/I2. The autophagy markers ATG4C and LC3II were upregulated after the low-concentration treatment of T-2 toxin and downregulated after the high-concentration treatment. After knocking down WISP1 expression in KBD chondrocytes, MAP1LC3B decreased while ATG4C and COL2A1 increased. Moreover, the rWISP1 protein treatment in C28/I2 chondrocytes could upregulate the expression of ATG4C and LC3II at the beginning and downregulate them then. Conclusions: Our study suggested that WISP1 might play a role in the pathogenesis of KBD through autophagy.

## 1. Introduction

Kashin-Beck disease (KBD) is a kind of endemic and chronic osteochondropathy, mainly distributed in the southwest and northeast regions of China [1]. Its early damage starts with the chondrocytes in the deep zone of cartilage. The main KBD manifestations include shortened fingers, enlarged joints, bone growth deformity or retardation, joint pain, second-osteoarthritis symptoms, etc. [2]. It mainly occurs during childhood or adolescence and becomes symptomatic with age.

A previous study showed that WISP1 might be a genetic control gene in KBD pathogenesis [3]. WISP1, also known as CCN4, has been found to get involved in the processes of cell differentiation, cell proliferation, tissue regeneration, and regulation of signal transduction [4]. It also plays an important role in the homeostasis, development, and recovery of bone and cartilage [4,5]. The expression of WISP1 was significantly increased in the articular cartilage of mice with experimental osteoarthritis (OA) [6]. A delayed wound healing and joint development can be seen in WISP1-knockout mice but not in the wildtype [4]. Moreover, WISP1 was found to be a trigger for the matrix-modulating enzymes released from chondrocytes, indicating its essential role in chondrocyte homeostasis [5]. WISP1 was also regarded as a potential therapeutic target for diseases related to the mammalian target of the rapamycin (mTOR) signaling pathway, the phosphatidylinositol-3-kinase (PI3-K) pathway, and the protein kinase B (Akt) pathway [7,8].

Defective autophagy was thought to play a role in the pathogenesis of KBD [9,10]. It was found that autophagy markers such as BECN1 and LC3 were dysregulated in KBD chondrocytes [10]. Furthermore, a two-stage genetic association study found that four single nucleotide polymorphisms (SNPs) in the ATG4C gene had significant association relationships with KBD, indicating the implication of ATG4C in the abnormal autophagy during KBD chondrocyte deterioration [9,10]. It was also reported that the autophagic response functions as a protective process in the terminal differentiated chondrocytes to antagonize cartilage damage [11]. Therefore, dysfunctional autophagy in KBD articular cartilage is possibly an important step in the development of KBD.

Cereal contamination by mycotoxin-producing fungi is thought to be a highly risky environmental factor for KBD [1]. T-2 toxin is the most explored among all the myco-toxins in the study of KBD etiology. In several studies, it was identified as being enriched in the cereal from KBD endemic areas or the KBD family [12,13]. Chicks fed with T-2 toxin-contaminated food showed similar cartilage changes to KBD, including the degeneration of knee joints, dysgenopathy, smaller and necrosis chondrocytes, etc. [14]. However, how the chondrocytes respond to the T-2 toxin and finally develop the KBD pathologic changes remains obscure.

In summary, the role of WISP1 in the pathogenesis of KBD is unclear. Therefore, it is hypothesized here that WISP1 may play a role in the deterioration of KBD chondrocytes triggered by T-2 toxin through the autophagy pathway. This study provides novel clues to understand the functional relevance and potential mechanism of WISP1 in KBD development.

## 2. Results

### 2.1. The Expression of WISP1 in KBD and Normal Control Articular Cartilage

Immunohistochemistry (IHC) was performed to detect the expression of WISP1 in articular cartilage from five matched pairs of KBD patients and controls. As the result showed, the positively stained cell number of WISP1 was found to be significantly higher in the middle zone of KBD cartilage compared with the control samples (Figure 1a, *p* = 0.0157). In addition, both the mRNA (Figure 1b, *p* = 0.0382) and the protein (Figure 1c, *p* = 0.0482) relative expression ratio of WISP1 in KBD chondrocytes exhibited an obvious increase when compared with control chondrocytes.

### 2.2. The Exploration of the Functional Role of WISP1 in KBD Chondrocyte Damage

Previous literature indicated that WISP1 can play a role in renal fibrosis through the autophagy pathway [15]. The relationship between WISP1 and the autophagy pathway was detected in this study. Firstly, we used cultured KBD chondrocytes to observe the ultrastructure changes under the TEM. Multiple changes were found in KBD chondrocytes, including the distorted nuclei, the swelling mitochondria, the expanded endoplasmic reticulum, and a great number of liposomes (Figure 2a). Moreover, the autolysosomes were found in the KBD chondrocytes (Figure 2a(i)), suggesting a potential participation of defective autophagy in the development of KBD, which had also been identified by other studies [9,10].

Then, we investigated the expression of autophagy-related genes (ATG4A/4C, ATG5, ATG7, BECN1, LC3A, LC3B) as well as chondrocyte phenotypic genes (COL2A1, ACAN). Paired chondrocyte samples harvested from five KBD patients and five controls with age and gender matching were used for qPCR and Western blot detection Appendix A). As shown in Figure 2b, no significant differences were found on the transcriptional scale. However, ATG4C, as an essential gene for autophagosome maturation, was found to be significantly up-regulated in KBD chondrocytes (*p* = 0.0019, Figure 2c,d).

Additionally, we further identified the role of WISP1 in KBD chondrocyte autophagy by siRNA interference based on KBD chondrocytes. KBD with the WISP1 lentivirus of the negative control (NC) and three targets (WISP1-A, WISP1-B, and WISP1-C) were used for the WISP1 knock-down cell model in KBD chondrocytes. As shown in Figure 2e,f, after 96 h of treatment, both lenti-WISP1-A (*p* = 0.0090) and lenti-WISP1-C (*p* = 0.0214) had a significant WISP1 gene knock-down effect on KBD chondrocytes. Finally, lenti-WISP1-A, with a better knock-down effect, was selected as a tool for the following experimentation. After incubating KBD chondrocytes (*n* = 3) with lenti-WISP1-A for 96 h, the mRNA expression of WISP1, ECM-related markers (COL2A1, ACAN, and SOX9), and autophagy-related genes (ATG4A/B/C, ATG5, ATG7, BECN1, and LC3A/B) was detected for WISP1 knock-down KBD chondrocytes and control KBD chondrocytes. As is shown in Figure 2g–i, the expression of WISP1 was significantly decreased both on mRNA (*p* = 0.0131) and protein scales (*p* = 0.0345) by lenti-WISP1-A interference treatment. Meanwhile, the mRNA (*p* = 0.0004) and protein expression (*p* = 0.0057) of COL2A1 were dramatically higher in the WISP1 knock-down KBD chondrocytes compared with KBD chondrocytes. For autophagy-related genes, both the mRNA (*p* = 0.0482) and protein (*p* = 0.0438) expression of ATG4C showed up-regulation in the WISP1 knock-down KBD chondrocytes. Meanwhile, the treatment of siRNA interference also lowered the expression of MAPLC3B (*p* = 0.0437).

### 2.3. Effects of T-2 Toxin Intervention on WISP1 Expression in C28/I2 Human Chondrocytes

Since the T-2 toxin is an important inducer of KBD, we also investigated how the WISP1 expression reacted to the T-2 toxin. Firstly, the MTT assay was applied to explore the effect of T-2 toxin on the cell viability of C28/I2 chondrocytes. The results showed that with the increase in T-2 toxin concentration, the cell survival rate of C28/I2 declined gradually. The median inhibitory concentrations (IC50) of T-2 toxin on C28/I2 chondrocytes were measured at 12 h, 24 h, and 72 h time points. Finally, the treatment concentrations of T-2 toxin (2, 5, and 8 ng/mL) and treatment period of 24 h were chosen for further experimentation (Appendix A).

TEM-captured images of C28/I2 chondrocytes with or without toxin treatment were also taken (Figure 3a). Autolysosomes were detected in the treatment groups of 2 ng/mL and 5 ng/mL, but not in the group of 8 ng/mL, which suggested the process of autophagy might be triggered by the lower concentrations of T-2 toxin (2 ng/mL and 5 ng/mL) (Figure 3a,c,d).

To investigate how the WISP1 and other molecular markers as described above react to the KBD risk factor (T-2 toxin), we also measured the expression levels of WISP1, ECM-related markers, and autophagy-related genes after the 24-h T-2 toxin treatment (2 ng/mL, 5 ng/mL, and 8 ng/mL). The results showed that WISP1 was induced by T-2 toxin. The expression of WISP1 showed an uptrend with the increase in T-2 toxin concentration. ATG4C and LC3II exhibited upregulation under low-concentration treatment (2 ng/mL, 5 ng/mL) and downregulation under high-concentration treatment (8 ng/mL), indicating autophagy might be activated first, then inhibited with the increase of WISP1 expression. Meanwhile, the amount of COL2A1 ECM marker decreased with increasing concentrations of T-2 toxin treatment (Figure 3b–d), indicating the degeneration of C28/I2 chondrocytes.

Unlike chondrocytes of KBD shown in Figure 2a, chondrocytes in Figure 3a have typical morphology after short-term exposure to T-2 toxin, but WISP1 as well as ECM-related markers and autophagy-related gene expression have significantly changed.

### 2.4. The Combined Effect of WISP1 and T-2 Toxin on Chondrocytes

To detect how WISP1 exerts its effect on chondrocyte damage with or without T-2 toxin, the rWISP1 treatment under different concentrations (5 ng/mL, 20 ng/mL, 100 ng/mL, and 500 ng/mL) was performed on the C28/I2 human chondrocytes. As shown in Figure 4, with the increasing concentration of rWISP1 (black line), a noticeable up-regulation of WISP1 was observed (*p* = 0.0021, Figure 4a). In addition, the mRNA expression of COL2A1 (*p* < 0.0001, Figure 4b) was down-regulated with the increasing expression of WISP1. Meanwhile, the mRNA expression of ATG4C (*p* = 0.0012), BECN1 (*p* < 0.0001), and MAP1LC3B (*p* = 0.0015) showed a significant uptrend with the increasing expression of WISP1 in low-concentration groups (below 100 ng/mL), but a downtrend in the high-concentration group (500 ng/mL), further confirming that autophagy might be activated under low WISP1 levels, then inhibited by the increasing expression of WISP1. However, the mRNA expression of ATG4B (*p* = 0.0005) decreased in all the groups, while ATG4A and ATG4C had the opposite trend in expression.

Moreover, the combined effect of the rWISP1 protein and T-2 toxin on C28/I2 was also explored (Figure 4). According to the results, the mRNA expression of WISP1 altered markedly under the rWISP1 protein treatment and had distinct trends in the three groups of T-2 toxin treatment (Figure 4a). The mRNA expression of ATG4C was also different in the three groups (Figure 4e). COL2A1 and ATG4A mRNA expression (Figure 4b,c) were significantly overexpressed and positively correlated to the increase in T-2 toxin concentration. In addition, ATG4B mRNA expression showed an overall downtrend in the three groups (Figure 4d). For BECN1 (Figure 4f) and MAP1LC3B (Figure 4g), a significant rise was observed in the three groups, suggesting a potential change in autophagy with an increasing concentration of T-2 toxin.

## 3. Discussion

In this study, we identified that WISP1 was involved in the pathogenesis of KBD chondrocytes by autophagy via constructing a T-2 toxin-treated chondrocyte cell model and conducting siRNA interference on KBD primary chondrocytes.

WISP1, whose full name is WNT1 inducible signaling pathway protein 1, has been known to be downstream in the WNT1 signaling pathway. Recently, WISP1 has gained more attention in bone and cartilage development and recovery. Bosch et al. found that WISP1 was highly expressed in experimental human OA cartilage and synovium [5]. However, the relationship between WISP1 and KBD is not known yet. Our previous study identified 123 genetic control genes of DNA methylation that may lead to the erosion of cartilage in KBD, among which the WISP1 gene was significantly up-regulated in KBD chondrocytes [3]. Therefore, in this study, we further detected the association between WISP1 and KBD development. Both the mRNA and protein expression of WISP1 were confirmed to be highly expressed in KBD chondrocytes. When knocking down WISP1 in KBD chondrocytes, the expression of COL2A1 was up-regulated, suggesting the potential therapy of WISP1 in chondrocyte recovery. Meanwhile, we also found some relationship between WISP1 and autophagy in KBD chondrocytes.

According to the results, WISP1 as well as ATG4C and LC3II have the same expression trend as the increasing concentration of T-2 toxin intervention, indicating the relationship between WISP1 and the ATG4/LC3II conjugation system [16]. ATG4C protein was significantly up-regulated in KBD chondrocytes, which was in accordance with the level of LC3II in T-2 toxin-treated C28/I2. Autophagy is a lysosomal degradation process that is vital for cell survival, differentiation, and development. ATG4 is the most important protease in the LC3 conjugation system [17]. Up to now, four ATG4 isoforms have been identified in mammals (ATG4A/B/C/D), but their specific mechanisms are still under debate [16]. A previous study indicated that ATG4B was the most active one [18]. The isoform followed is ATG4A, while ATG4C and ATG4D had minimal activities in vitro [18]. BECN1 and LC3 are the critical markers, initiators, and regulators of autophagy [19]. LC3 is known to exist in two forms (LC3I and LC3II). LC3I is mainly found in the cytoplasm. LC3II, converted from LC3I, exists in both the inner and outer membranes of the autophagosome and is responsible for the identification of autophagosomal membranes. Autophagy can be activated by related protein complexes that initiate the sequestration of cytoplasmic parts into double-membrane vacuoles, the autophagosomes [20,21]. The fusion of the autophagosomes with lysosomes is a key step in autophagy. After the fusion, intra-autophagosomal LC3II is degraded by lysosomal proteases [22]. Therefore, changes in cellular LC3II levels are associated with the dynamic turnover of LC3II and also as a marker of autophagic activity [22]. We found that the ATG4/LC3II conjugation system was activated at low WISP1 levels and inhibited when WISP1 rose to a high level (Figure 3 and Figure 4). It could partly explain the different expression of ATG4C in KBD chondrocytes between our study and a previous one, in which ATG4C was found to be decreased in KBD cartilage [9]. The bidirectional results of the abnormal autophagy level were detected in KBD chondrocytes [9,10,23], because the effect of WISP1 on the autophagy of chondrocytes is dynamic.

Changes in cartilage ECM were also observed in this study. It is known that during chondrogenesis, mature chondrocytes can secrete collagen II, proteoglycans, and non-collagenous proteins to form the ECM. Previous studies showed that T-2 toxin could decrease type II collagen [24], which has also been confirmed by this study. With the T-2 toxin concentration increasing, COL2A1 decreased gradually while WISP1 increased gradually. Overexpression of WISP1 was known to be able to regulate cartilage matrix production in human OA [23,25]. In this study, collagen II, the major cartilage ECM component and marker of chondrocytes, was also found to be regulated by WISP1. The expression of COL2A1 decreased under the treatment of rWISP1 (below 100 ng/mL) and slightly increased when rWISP1 was above 500 ng/mL. This indicated the potential relationship between WISP1 and COL2A1. After knocking down WISP1 in KBD chondrocytes, COL2A1 was observed to increase significantly, which further confirmed the role of WISP1 in regulating cartilage matrix production.

There are several limitations to this study. Firstly, when cultured in vitro, the chondrocyte phenotype may change. It is necessary to perform in vivo studies to verify the findings in the future. Secondly, due to the difficulty in collecting age-matched knee cartilage as controls of KBD cartilage, we adopted a compromising design in which cartilage at different anatomical regions (femoral neck) was used for comparisons. This might have some influence on our results. Thirdly, since we had not obtained enough control cartilages due to the ethical issue, we did not provide the TEM images of control chondrocytes for comparison. Previously published TEM images of normal chondrocytes were collected for comparison and to capture the features of KBD chondrocytes [26]. Moreover, the sample size used in this study was relatively small. There were only five KBD and five control samples due to the difficulty of collecting human cartilage samples. Although we had found an overall overexpression of WISP1 in KBD compared with controls, one KBD sample did not appear unregulated. Moreover, we also observed weak expressions of BECN1, ATG4C, and ATG4A in this sample. Therefore, large KBD samples and multilevel studies, including in vitro and in vivo, are still needed to validate the results.

## 4. Materials and Methods

### 4.1. Ethical Statement

This research was approved by the Human Ethics Committee of Xi’an Jiaotong University in 2015 (Project Number: 2015-237). The study was conducted in accordance with ethical standards as laid down in the 1964 Declaration of Helsinki and its later amendments or comparable ethical standards. A written informed consent was signed by all the participants or their relatives.

### 4.2. Specimen Collection

KBD was diagnosed according to the diagnosis criteria for KBD in China (WS/T 207-2010). In this study, human cartilage samples were collected from nine KBD patients with grade III KBD and eight control subjects (Appendix A). Specimens of cartilage from KBD patients were harvested from the same anatomic spot of knee femoral condyles, while control samples were collected from total hip arthroplasty and hemiarthroplasty for femoral neck fractures. Patients with osteoporosis, rheumatoid arthritis, primary OA, and other bone and cartilage diseases were excluded from this study. The C28/I2 human chondrocyte cell line was donated by Ms. Marry B. Goldring from Weill Cornell Medical College [27].

### 4.3. Chondrocyte Culture

Cartilages from KBD and control were isolated from chondrocytes within 10 h after the surgical operation. Cartilage samples were cut into 5–10 mm^3^ slices, treated with 0.25% trypsin (Xi’an GuoAn Biological Techonology Company, Xi’an, China) for 30 min at RT, and immersed into 0.2% type II collagenase solution (Gibco, 17101-015, CarIsbad, ON, Canada) at 37 °C for 10 h. After that, the isolated chondrocytes were cultured at 37 °C in 5% CO_2_ in DMEM/F12 supplemented with 10% fetal calf serum (Sijiqing, Zhejiang Tianhang Biotechnology Company, Hangzhou, China) and 1% penicillin/streptomycin (HyClone, Logan, UT, USA). In this study, the passage for the first generation of human chondrocytes was used. The procedure for culturing C28/I2 chondrocytes was the same.

### 4.4. Tissue Preparation and Hematoxylin-Eosin (H&E) Staining

The human cartilage samples were cut into 30 × 30 mm^2^, 1–2 cm slices, and embedded in 4% paraformaldehyde within 10 h after the surgical operation. A total of 18% EDTANa2 was used for bone demineralization after cutting the slices to 0.5 mm. Then, the sections were dehydrated, cleared, and embedded in liquid paraffin. After that, the sections were cut into 5 μm slices before the next experiment. For H&E staining, the slices were dewaxed and dehydrated before hematoxylin (8 min) and eosin staining (2 min).

### 4.5. IHC

The prepared cartilage slices were subjected to dewaxing, rehydration, and antigen retrieval by using citrate buffer (pH 6.0) overnight at 37 °C and 12.5% trypsin (Xi’an GuoAn Biological Technology Company, Xi’an, China.) at 37 °C for 20–30 min. The following process was performed using Rabbit SP reagent (Beijing Zhong Shan Gold Bridge Biological Technology Company, Beijing, China.) according to the manual. After incubating with the primary antibody of WISP1 (1:60, GTX110512, Gene Tex Inc., San Antonio, TX, USA) and the secondary antibody of Rabbit SP reagent, DAB solution was used to treat the sections until the proper color could be seen.

### 4.6. Morphology Observation by Transmission Electron Microscope (TEM)

KBD chondrocytes as well as C28/I2 human chondrocytes with or without T-2 toxin were observed under TEM. Briefly, 2.5% glutaraldehyde was used to fix the chondrocytes at 4 °C for over 2 h. Then, the sample was washed by 0.1 M PBS for 30 min, fixed with 1% osmic acid at 4 °C for 2 h, and washed again by 0.1M PBS for 10 min. After that, the sample was dehydrated by alcohol, exchanged by propylene oxide, and embedded in epoxy resin before being cut into 50–70 nm sections and observed by TEM (Hitachi 7650, 80 KV, Tokyo, Japan).

### 4.7. Cell Viability Detection by the Methyl-Thiazol-Tetrazolium (MTT) Assay

T-2 toxin (J&K Scientific Ltd., Beijing, China) was used to make the KBD cell models in C28/I2 chondrocytes as described previously [13,28]. The cytotoxicity was evaluated by MTT assay after exposing C28/I2 to T-2 toxin. Conditions of different T-2 concentrations (1, 2, 5, 8, 10 ng/mL) and times (12 h, 24 h, 72 h) were set to treat C28/I2 cells in 96-well plates. Each condition has five repeats.

### 4.8. Intervention of rWISP1 Protein in C28/I2 Human Chondrocytes

Four concentrations of rWISP1 protein (Peprotech 120-18) were used here to intervene C28/I2 chondrocytes for 24 h (5, 20, 100, 500 ng/mL) in 6-well plates [5,29].

### 4.9. WISP1 Knock-Down Cell Model by Lentiviral Transduction

Lentiviral transduction was conducted based on three KBD patients using the recombinant lentivirus WISP1 constructed by Genechem (Shanghai, China). The transduction was made under 100 multiplicities of infection (MOI), according to the pre-experiment results. Chondrocytes were infected by lentivirus for 12 h and cultured in the medium of DMEM/F12 supplemented with 10% fetal calf serum for 96 h. The efficiency of infection was measured to be around 80% by the fluorescence inverse microscope, then the medium was discarded and TRIzol was added to end the infection.

### 4.10. mRNA Detection by qPCR

Total RNA was extracted from chondrocytes according to the TRIzol reagent protocol (Invitrogen Ltd., CarIsbad, ON, Canada) and reverse-transcribed to cDNA by a one-step kit (TaKaRa PrimeScriptTM, Beijing, China). The mRNA expression of selected genes and GAPDH was measured by qPCR with triplicate analysis (Appendix A).

### 4.11. Protein Detection by Western Blot

RIPA reagent (Xi’an Hat Biotechnology Co., Ltd., Xi’an, China) and TRIzol were used to extract protein from chondrocytes in this study. The protein suspension was equally added into the well of the SDS-polyacrylamid gelectrophoresis (SDS-PAGE) for separation, transferred to polyvinylidenedifluoride (PVDF) membranes, and blocked either in 5% nonfat milk (for total proteins) or in 3% FBS (for phosphorylated proteins) solution before being incubated with primary antibodies at 4 °C overnight, such as WISP1 (1:2000, GTX110512, Gene Tex, San Antonio, TX, USA), COL2A1 (1:1000, ab188570, abcam, Cambridge, UK), BECN1 (1:2000, ab207612, abcam), ATG4C (1:1000, ab183516, abcam), ATG4A (1:1000, ab108322, abcam), LC3B (1:2000, ab192890, abcam), and GAPDH (1:10,000, 10494-1-AP, proteintech, Wuhan, China), and the secondary antibody (1:5000, Xi’an Zhuangzhi Biotechnology Co., Ltd., Xi’an, China) for 1 h at RT. Gene Gnome XRQ (Gene Co., Ltd., Hongkong, China) was used to analyze the protein bands.

### 4.12. Statistical Analysis

A one-way analysis of variance (ANOVA) and an independent sample *t*-test were used for comparisons of the results. Statistical analyses were performed using SPSS 23.0. The threshold was defined as *p <* 0.05.

## 5. Conclusions

This study revealed the important role of WISP1 in the abnormal autophagy of KBD pathogenesis. On one hand, the lower expression of WISP1 would activate the autophagy process in chondrocytes, especially the ATG4/LC3II conjugation system. On the other hand, the higher expression of WISP1 would inhibit the autophagy process in chondrocytes. It indicated a bidirectional and dynamic effect of WISP1 on the autophagy of chondrocytes. Furthermore, WISP1 could regulate the expression of cartilage matrix production (COL2A1) in KBD chondrocytes. In conclusion, this study suggested that WISP1 might play a vital role in the pathogenesis of KBD via the ATG4C/LC3II autophagy process.

## Figures and Tables

**Figure 1 ijms-24-16037-f001:**
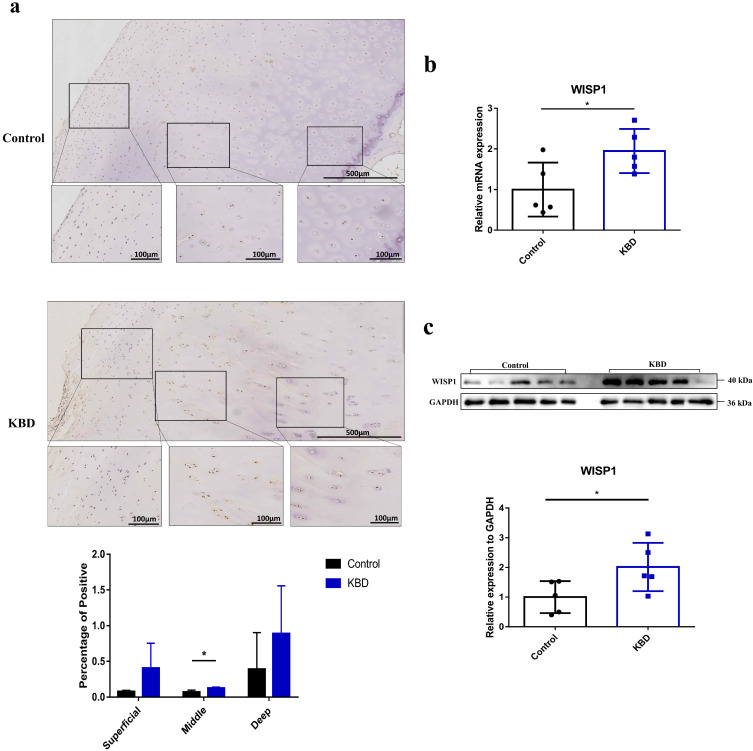
The expression of WISP1 in KBD and control chondrocytes. (**a**) The scale bars are 500 μm and 100 μm individually. * *p* < 0.05. (**b**) The mRNA expression of WISP1 in KBD and control chondrocytes. * *p* < 0.05. *n* = 5. (**c**) The protein expression of WISP1 in KBD and control chondrocytes. * *p* < 0.05. *n* = 5.

**Figure 2 ijms-24-16037-f002:**
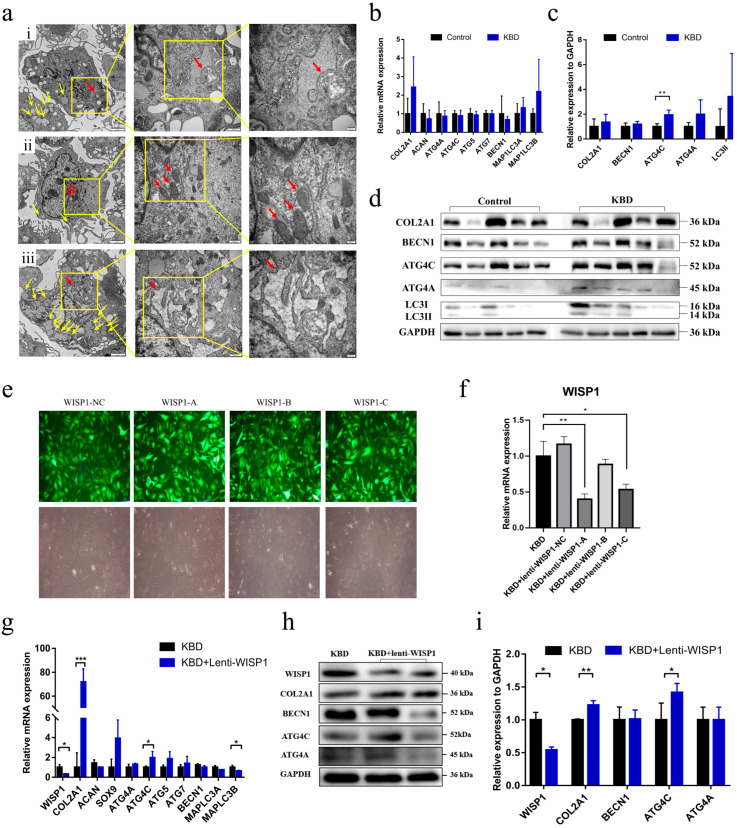
The functional role of WISP1 in KBD chondrocyte damage. (**a**) The TEM results of KBD chondrocytes. Red arrows represent (**a**(i)) The autolysosome in KBD chondrocyte, (**a**(ii)) The mitochondria in KBD chondrocyte, and (**a**(iii)) The endoplasmic reticulum in KBD chondrocyte. The yellow arrows represent a great number of liposomes. The scale bars are 2 μm, 500 nm, and 200 nm, respectively, from left to right. (**b**) The mRNA expression of selected genes in KBD and control chondrocytes. *n* = 5. (**c**) The protein expression of markers in KBD and controls. *n* = 5. (**d**) The Western blot results of markers in KBD and controls. (**e**) The bright field and dark field of the efficiency of three targets of the WISP1 lentivirus in KBD chondrocytes. The magnification is 100×. (**f**) The mRNA expression of WISP1 in KBD chondrocytes and KBD with the WISP1 lentivirus of the negative control (NC) and three targets (WISP1-A, WISP1-B, and WISP1-C). (**g**) The mRNA expression of WISP1 and other genes in controls, KBD chondrocytes, and KBD of the WISP1-A lentivirus. *n* = 3. (**h**) The Western blot result from markers in WISP1 knock-down KBD chondrocytes and KBD chondrocytes. (**i**) The protein expression of WISP1 and other markers in WISP1 knock-down KBD chondrocytes and KBD chondrocytes. *n* = 3. * *p* < 0.05, ** *p* < 0.01, *** *p* < 0.001.

**Figure 3 ijms-24-16037-f003:**
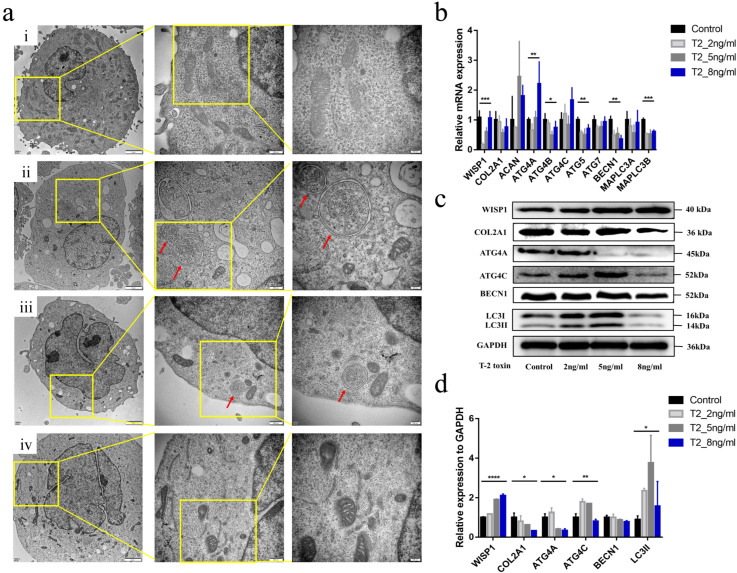
Effects of T-2 toxin intervention in C28/I2 human chondrocytes. (**a**) The TEM results of C28/I2 human chondrocytes. (**a**(i)) The mitochondria in control C28/I2 human chondrocytes; (**a**(ii)) The autolysosome (red arrows) in C28/I2 chondrocytes treated with T-2 toxin (2 ng/mL); (**a**(iii)) The autolysosome (red arrows) in C28/I2 chondrocytes treated with T-2 toxin (5 ng/mL); (**a**(iv)) The mitochondria in C28/I2 chondrocytes treated with T-2 toxin (8 ng/mL). The scale bars are 2 μm, 500 nm, and 200 nm, respectively, from left to right. (**b**) The mRNA expression of target genes in C28/I2 intervened by T-2 toxin. (**c**) The protein expression of target markers in C28/I2 intervened by T-2 toxin. (**d**) The results of Western blots in C28/I2 intervened by T-2 toxin. * *p* < 0.05, ** *p* < 0.01, *** *p* < 0.001, **** *p* < 0.0001.

**Figure 4 ijms-24-16037-f004:**
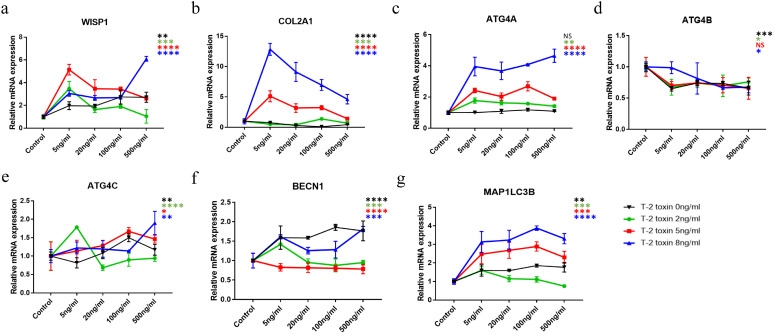
The combined effect of rWISP1 with or without T-2 toxin in C28/I2 human chondrocytes. The mRNA expression of *WISP1* (**a**), *COL2A1* (**b**), *ATG4A* (**c**), *ATG4B* (**d**), *ATG4C* (**e**), *BECN1* (**f**), and *MAP1LC3B* (**g**) in C28/I2 was influenced by the four different concentrations of rWISP1 protein with or without three different concentrations of T-2 toxin. The abscissa stands for the four different concentrations of rWISP1 protein (5, 20, 100, and 500 ng/mL). The black, green, red, and blue lines stand for 0, 2, 5, and 8 ng/mL of T-2 toxin separately. NS stands for no sense. * *p* < 0.05, ** *p* < 0.01, *** *p* < 0.001, **** *p* < 0.0001.

## Data Availability

The datasets used and/or analyzed during the current study are available from the corresponding author on reasonable request.

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
