# Peer review of "WISP1 Is Involved in the Pathogenesis of Kashin-Beck Disease via the Autophagy Pathway"

_ijms, 2023, doi:10.3390/ijms242216037_

Round 1

Reviewer 1 Report

Comments and Suggestions for Authors

The authors investigated the functional relevance and potential mechanism of Wnt-inducible signalling pathway protein 1 (WISP1) in the pathogenesis of Kashin-Beck disease. They carried out morphological and histopathological observations under light and electron microscope on diseased and control cartilage samples. The expression levels of WISP1 and molecular markers related to autophagy pathway and extracellular matrix synthesis were detected in cultured sick and control chondrocytes by qRT-PCR, Western blot, and immunohistochemistry. Furthermore, lentiviral transfection technique was applied to make a WISP1 knockdown cell model based on Kashin-Beck diseased chondrocytes. In vitro intervention experiments were also conducted on C28/I2 human chondrocytes cell line using human recombinant WISP1. They found that WISP1 seems to play a role in the pathogenesis of Kashin-Beck disease via the autophagy process.

First and most importantly, authors are recommended to overview their usage of terms involved in autophagy process. Please, refer to the cited papers (17. Zheng K et al. 2016 and 18. Levine B and Kroemer G. 2008) which use the expressions in the right way. For example, please, change ‘autophagy-lysosome’ to ‘autolysosome’ at all the 11 occurrences except those below.

lines 240-242 ‘It can be activated by the combination of multiple autophagy-related proteins by forming the double-membrane vacuoles, the autophagy-lysosome’ change to ‘It can be activated by autophagy-related protein complexes which initiate the sequestration of cytoplasmic parts into double-membrane vacuoles, the autophagosomes’

line 242 ‘The fusion of the autophagy-lysosome is a key final step of autophagy.’ change to ‘The fusion of the autophagosomes with lysosomes is a key step of autophagy.’

lines 245-246 ‘the internal and external compartment of the autophagosome and is responsible to initiate the formation’ change to ‘the inner and outer membranes of the autophagosome and is responsible for the identification’

lines 247-248 ‘During the fusion of autophagy-lysosome, intra-autophagosomal LC3II is degraded by lysosomal proteases’ change to ‘After the fusion of autophagosomes with lysosomes, intra-autophagosomal LC3II is degraded by lysosomal proteases’

line 320 (4.4 Tissue preparation and hematoxylin-eosin (H&E) staining) begins with a missing part, please check what has been left out.

line 336 Please, use the proper chemical name of ‘glutaraldehyde’ instead of ‘glutaral’.

line 339 Please, use ‘propylene oxide’ instead of ‘epoxy propane’.

line 340 Please, give details on the TEM used (manufacturer and type e.g. Jeol TEM-1011) and state the acceleration voltage applied (e.g. 80 kV).

Comments on the Quality of English Language

The present manuscript needs some English language editing. Please, correct the following:

line 21 ‘under the optical and electron microscope’ change to ‘under light and electron microscope’

line 25 ‘make the WISP1 knockdown cell model’ change to ‘make a WISP1 knockdown cell model’

lines 35-36 and 226 ‘via ATG4C/LC3II 35 autophagy process’ change to ‘by autophagy’

line 110 ‘the autophagy-lysosome was found’ change to ‘autolysosomes were found’

line 111 ‘participation of disturbed autophagy’ change to ‘participation of defective autophagy’

line 118 ‘autophagosome mature,’ change to ‘autophagosome maturation,’

line 121 ‘by the way of siRNA interference’ change to ‘by siRNA interference’

lines 164-165 ‘The TEM-captured images of C28/I2 chondrocytes with or without toxin treatment 164 was also detected’ change to ‘TEM-captured images of C28/I2 chondrocytes with or without toxin treatment were also taken’

line 166 ‘To the results, autophagy-lysosome can be detected’ change to ‘Autolysosomes were detected’

line 168 ‘triggered by the low concentration of T-2’ change to ‘triggered by the lower concentrations of T-2’

lines 178-179 ‘the ECM relative marker of COL2A1, was identified downtrend as the increasing concentration of T-2 toxin treatment’ change to ‘the amount of COL2A1 ECM marker decreased with increasing concentration of T-2 toxin treatment’

lines 213-214 ‘change of autophagy as the increasing concentration’ change to ‘change of autophagy with increasing concentration’

line 227 ‘was significant up-regulated’ change to ‘was significantly up-regulated’

line 228 ‘could induce the uptrend expression’ change to ‘could induce the increasing expression’

line 229 ‘LC3II protein were increased’ change to ‘LC3II proteins was increased’

line 238 ‘seldomly observed’ change to ‘rarely observed’

line 262 ‘It was suggested ATG4/LC3II conjugation system was activated’ change to ‘We found that ATG4/LC3II conjugation system was activated’

line 340 ‘slices’ change to ‘sections’

Author Response

Reviewer 1

  1. Comments: First and most importantly, authors are recommended to overview their usage of terms involved in autophagy process. Please, refer to the cited papers (17. Zheng K et al. 2016 and 18. Levine B and Kroemer G. 2008) which use the expressions in the right way. For example, please, change ‘autophagy-lysosome’ to ‘autolysosome’ at all the 11 occurrences except those below.

Response: Thank you for your comments!

We appreciate your time and patience to help improve the language of this article. Per your guidance, we have changed “autophagy-lysosome” to “autolysosome” at all the 11 occurrences in the manuscript. Please see them in the revised manuscript.

Thanks again!

  1. Comments: lines 240-242 ‘It can be activated by the combination of multiple autophagy-related proteins by forming the double-membrane vacuoles, the autophagy-lysosome’ change to ‘It can be activated by autophagy-related protein complexes which initiate the sequestration of cytoplasmic parts into double-membrane vacuoles, the autophagosomes’

Response: Thank you for your comments!

Per your guidance, we have changed “It can be activated by the combination of multiple autophagy-related proteins by forming the double-membrane vacuoles, the autophagy-lysosome” to “Autophagy can be activated by related protein complexes which initiate the sequestration of cytoplasmic parts into double-membrane vacuoles, the autophagosomes”. Please see lines 240-241 of the revised manuscript.

Thanks again!

  1. Comments: line 242 ‘The fusion of the autophagy-lysosome is a key final step of autophagy.’ change to ‘The fusion of the autophagosomes with lysosomes is a key step of autophagy.’

Response: Thank you for your comments.

Per your guidance, we have changed “The fusion of the autophagy-lysosome is a key final step of autophagy.” to “The fusion of the autophagosomes with lysosomes is a key step of autophagy”. Please see lines 241-242 of the revised manuscript.

Thanks again!

  1. Comments: lines 245-246 ‘the internal and external compartment of the autophagosome and is responsible to initiate the formation’ change to ‘the inner and outer membranes of the autophagosome and is responsible for the identification’

Response: Thank you for your comments!

Per your guidance, we have changed “the internal and external compartment of the autophagosome and is responsible to initiate the formation” to “the inner and outer membranes of the autophagosome and is responsible for the identification”. Please see lines 238-239 of the revised manuscript.

Thank again!

  1. Comments: lines 247-248 ‘During the fusion of autophagy-lysosome, intra-autophagosomal LC3II is degraded by lysosomal proteases’ change to ‘After the fusion of autophagosomes with lysosomes, intra-autophagosomal LC3II is degraded by lysosomal proteases’

Response: Thank you for your comments!

Per your guidance, we have changed “During the fusion of autophagy-lysosome, intra-autophagosomal LC3II is degraded by lysosomal proteases” to “After the fusion, intra-autophagosomal LC3II is degraded by lysosomal proteases”. Please find them in lines 242-243 of the revised manuscript.

Thanks again!

  1. Comments: Line 320 (4.4 Tissue preparation and hematoxylin-eosin (H&E) staining) begins with a missing part, please check what has been left out.

Response: Thank you for your comments!

We are sorry for missing some information here. Per your guidance, we have added the missing sentence in the beginning. “The human cartilage samples were cut into 30×30 mm2, 1-2 cm slices and embedded in 4% paraformaldehyde within 10 hours after the surgical operation.” Please see line 309 of the revised manuscript.

Thanks again!

  1. Comments: line 336 Please, use the proper chemical name of ‘glutaraldehyde’ instead of ‘glutaral’.

Response: Thank you for your comments!

We are sorry for using the inaccurate chemical term. Per your guidance, we have changed “glutaral” to “glutaraldehyde”. Please see line 325 of the revised manuscript.

Thanks again!

  1. Comments: line 339 Please, use ‘propylene oxide’ instead of ‘epoxy propane’.

Response: Thank you for your comments!

We are sorry again for using the inaccurate chemical term. Per your guidance, we have changed “epoxy propane” to “propylene oxide”. Please see line 328 of the revised manuscript.

Thanks again!

  1. Comments: line 340 Please, give details on the TEM used (manufacturer and type e.g. Jeol TEM-1011) and state the acceleration voltage applied (e.g. 80 kV).

Response: Thank you for your comments.

We are sorry for missing this information here. The manufacturer of TEM we used is Hitachi 7650, and the acceleration voltage applied is 80kV. Per your guidance, we have added such information, Hitachi 7650 and 80KV, into the revised manuscript. Please see line 329 of the revised manuscript.

Thanks again!

  1. Comments: Comments on the Quality of English Language

The present manuscript needs some English language editing. Please, correct the following:

line 21 ‘under the optical and electron microscope’ change to ‘under light and electron microscope’

line 25 ‘make the WISP1 knockdown cell model’ change to ‘make a WISP1 knockdown cell model’

lines 35-36 and 226 ‘via ATG4C/LC3II 35 autophagy process’ change to ‘by autophagy’

line 110 ‘the autophagy-lysosome was found’ change to ‘autolysosomes were found’

line 111 ‘participation of disturbed autophagy’ change to ‘participation of defective autophagy’

line 118 ‘autophagosome mature,’ change to ‘autophagosome maturation,’

line 121 ‘by the way of siRNA interference’ change to ‘by siRNA interference’

lines 164-165 ‘The TEM-captured images of C28/I2 chondrocytes with or without toxin treatment 164 was also detected’ change to ‘TEM-captured images of C28/I2 chondrocytes with or without toxin treatment were also taken’

line 166 ‘To the results, autophagy-lysosome can be detected’ change to ‘Autolysosomes were detected’

line 168 ‘triggered by the low concentration of T-2’ change to ‘triggered by the lower concentrations of T-2’

lines 178-179 ‘the ECM relative marker of COL2A1, was identified downtrend as the increasing concentration of T-2 toxin treatment’ change to ‘the amount of COL2A1 ECM marker decreased with increasing concentration of T-2 toxin treatment’

lines 213-214 ‘change of autophagy as the increasing concentration’ change to ‘change of autophagy with increasing concentration’

line 227 ‘was significant up-regulated’ change to ‘was significantly up-regulated’

line 228 ‘could induce the uptrend expression’ change to ‘could induce the increasing expression’

line 229 ‘LC3II protein were increased’ change to ‘LC3II proteins was increased’

line 238 ‘seldomly observed’ change to ‘rarely observed’

line 262 ‘It was suggested ATG4/LC3II conjugation system was activated’ change to ‘We found that ATG4/LC3II conjugation system was activated’

line 340 ‘slices’ change to ‘sections’

Response: Thank you for your comments!

We really appreciate all your efforts to help us improve the language from every detail. Per your guidance, we have corrected all the mistakes you listed above. Please see them in the revised manuscript.

In line 19-20, ‘under the optical and electron microscope’ has been changed to ‘under light and electron microscope’.

In line 23, ‘make the WISP1 knockdown cell model’ has been changed to ‘make a WISP1 knockdown cell model’.

In lines 33 and 211, ‘via ATG4C/LC3II autophagy process’ has been changed to ‘by autophagy’.

In lines 96-97, ‘the autophagy-lysosome was found’ has been changed to ‘autolysosomes were found’.

In lines 97-98, ‘participation of disturbed autophagy’ has been changed to ‘participation of defective autophagy’.

In line 105, ‘autophagosome mature,’ has been changed to ‘autophagosome maturation,’.

In line 108, ‘by the way of siRNA interference’ has been changed to ‘by siRNA interference’.

In lines 164-165, ‘The TEM-captured images of C28/I2 chondrocytes with or without toxin treatment was also detected’ has been changed to ‘TEM-captured images of C28/I2 chondrocytes with or without toxin treatment were also taken’.

In line 151, ‘To the results, autophagy-lysosome can be detected’ has been changed to ‘Autolysosomes were detected’.

In line 153, ‘triggered by the low concentration of T-2’ has been changed to ‘triggered by the lower concentrations of T-2’.

In lines 162-163, ‘the ECM relative marker of COL2A1, was identified downtrend as the increasing concentration of T-2 toxin treatment’ has been changed to ‘the amount of COL2A1 ECM marker decreased with increasing concentration of T-2 toxin treatment’.

In lines 199-200, ‘change of autophagy as the increasing concentration’ has been changed to ‘change of autophagy with increasing concentration’.

In lines 229, ‘was significant up-regulated’ has been changed to ‘was significantly up-regulated’.

In lines 245-246, ‘It was suggested ATG4/LC3II conjugation system was activated’ has been changed to ‘We found that ATG4/LC3II conjugation system was activated’.

In line 329, ‘slices’ has been change to ‘sections’.

Thanks again!

Reviewer 2 Report

Comments and Suggestions for Authors

In the current study, the authors evaluated the involvement of the WISP1 pathway in the pathogenesis of Kashin-Beck disease. The authors suggest that WISP1 might play a role in the pathogenesis of KBD via activation of the ATG4C/LC3II autophagy process.

This is an interesting study, nevertheless it required significant improvement. Some of the described results are not substantiated by the data presented in the figures. The Discussion needs to be re-written. Please see the specific comments below.

Abstract:

Line 30: “The autophagy makers” Markers?

Results:

Figure 1a. It is impossible to see the stained cells at this magnification. Please provide images of high magnification in addition to the low magnification.

Line 96: “cell necrosis, appearing as the blue dye in and around chondrocytes”. The blue dye is hematoxylin which stains DNA and RNA. It is not an indicator of necrosis. I do not see any clear differences between KBD and control at this magnification in Additional file 1.

Line 98: “The erosion also could be seen in the surface of cartilage.”

You should not make statements that cannot be confirmed when looking at the Additional file 1 pictures. If you are convinced that you observed something and can prove it, then you should include the Figure into the main manuscript file and show high magnification inserts with arrows pointing to the changes that you state are taking place. Otherwise, remove Section 2.2 from the manuscript.

Line 106: “we observed the morphological changes of KBD chondrocytes under the TEM”. You should describe that these were cultured chondrocytes and for how long they were cultured. From your description it is unclear are these chondrocytes from the tissue or from culture.

Figure 2A. Where is a comparison with the morphology of the control cultured chondrocytes? How do you know that KBD chondrocytes have “distorted nuclei, the swelling mitochondria, the expanded endoplasmic reticulum, and a great number of liposomes”. Moreover, the chondrocytes in Figure 3A have typical morphology with smooth borders. The presumable chondrocytes in Figure 2a look like typical macrophages with long processes. What markers were used to verify that the majority of cells that you isolated from cartilage, cultured and analyzed were chondrocytes and not macrophages/other cell types?

Discussion

Lines 224-238: You do not have to repeat the results that just were described in the Results section. You need to discuss your data comparing it with the previously published studies and providing your interpretation of the results.

Lines 239-249: This is description of the autophagy process. There is no discussion/comparison with your data.

The entire Discussion section need to be re-written.

Methods:

Line 295: Please provide the date of approval.

Line 304: “Specimens of cartilage from KBD patients were harvested from the same anatomic spot of knee femoral condyles while control samples were collected from total hip arthroplasty and hemiarthroplasty for femoral neck fractures”. Does it mean that KBD and control cartilage samples were collected from different anatomical regions? What effect this could have had on the results of your experiments?

Comments on the Quality of English Language

The language of the manuscript requires significant improvement. Many sentences are awkwardly structured and difficult to read and understand. 

For example: 

Line 115: “Five KBD and five matched control chondrocytes were enrolled for qPCR and western-blot experiments”.?
